# Unsupervised Sounding Pixel Learning

Yining Zhang[1], Yanli Ji *[1,2] and Yang Yang[1,3]

[1]School of Computer Science and Engineering, UESTC
[2]Shenzhen Institute for Advanced Study, UESTC
[3]Institute of Electronic and Information Engineering of UESTC in Guangdong, China
zyn@std.uestc.edu.cn, {yanliji,yang.yang}@uestc.edu.cn.

## Abstract

Sounding source localization is a challenging cross-modal task due to the difficulty of cross-modal alignment. Although supervised cross-modal methods achieve encouraging performance, heavy manual annotations are expensive and inefficient. Thus it is valuable and meaningful to develop unsupervised solutions. In this paper, we propose an **U**nsupervised **S**ounding **P**ixel **L**earning (USPL) approach which enables a pixel-level sounding source localization in unsupervised paradigm. We first design a mask augmentation based multi-instance contrastive learning to realize unsupervised cross-modal coarse localization, which aligns audio-visual features to obtain coarse sounding maps. Secondly, we present an *Unsupervised Sounding Map Refinement (SMR)* module which employs the visual semantic affinity learning to explore inter-pixel relations of adjacent coordinate features. It contributes to recovering the boundary of coarse sounding maps and obtaining fine sounding maps. Finally, a *Sounding Pixel Segmentation (SPS)* module is presented to realize audio-supervised semantic segmentation. Extensive experiments are performed on the AVSBench-S4 and VG-GSound datasets, exhibiting encouraging results compared with previous SOTA methods.

## 1 Introduction

Audio-visual localization tasks have attracted much attention in recent years. Various works have made great achievements, including Sound Source Localization (SSL)(Chen et al., 2021; Mo and Morgado, 2022b), Audio-Visual Event localization (AVE), Audio-Visual Video Parsing (AVVP)(Wu and Yang, 2021; Lin et al., 2021), and Audio-Visual Segmentation (AVS)(Zhou et al., 2022). In this work, we focus on a variant of SSL task : unsupervised pixel-level sound source localization (pixel SSL), which aims at accurately localizing the sounding objects

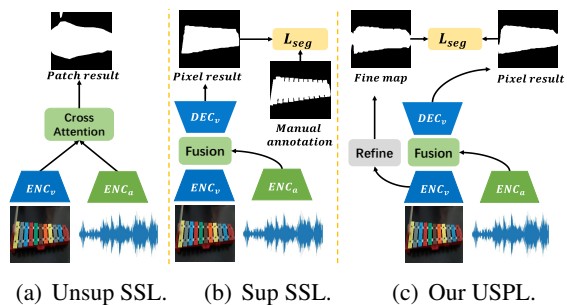

(a) Unsup SSL.  (b) Sup SSL.  (c) Our USPL.

Figure 1: **Comparison of our USPL with previous SSL tasks.** (a) Unsupervised Sound Source Localization (Unsup SSL)(Chen et al., 2021) calculates audio-visual cross attention to obtain patch-level coarse results. (b) Supervised Sound Source Localization (Sup SSL)(Zhou et al., 2022) provides a supervised label for model training. (c) Our Unsupervised Sounding Pixel Learning (USPL) aims to obtain pixel-level localization results in an unsupervised paradigm.

in a given audio-visual scene without any annotations.

Despite recent advancements in SSL, some challenges remain to be addressed. One of the primary challenges is to learn accurate pixel-level sounding maps rather than approximate patch-level maps. As shown in Fig.1(a), Unsup SSL methods (Chen et al., 2021) always learn the audio-visual co-occurrence by contrastive learning (Chen et al., 2020b), then directly calculate the audio-visual attention to estimate the sounding maps. However, these approaches often output coarse and unreliable-sounding masks containing only localization information but lack object shape information.

Another challenge is doing fine-grained sound source localization in a totally unsupervised paradigm. As depicted in Fig.1(b), to precisely identify regions of sounding, Sup SSL methods (Zhou et al., 2022) build a pixel-level labeled dataset and utilize semantic segmentation models to produce pixel-level results. However, Sup SSL

methods are limited by the requirement for intensive annotations, which can be a burdensome task and hard to be feasible in real environments. Therefore, in Unsupervised Sounding Pixel Learning (USPL, Fig. 1(c)), our motivation is to produce precise pixel-level sounding object maps in videos by a totally unsupervised paradigm.

To address the challenge of pixel SSL, we first conduct mask augmentation based multi-instance contrastive learning (MMICL) to align the paired audio-visual features. We highlight that the mask augmentation layer greatly alleviates the overfit and sub-optimization problem proposed in previous work(Chen et al., 2021; Mo and Morgado, 2022b). After completing the MMICL, we can use audio-visual attention to obtain patch-level sounding object results which are considered as coarse sounding maps.

However, these coarse sounding maps fall short of our pixel SSL requirements. Therefore, we perform an unsupervised Sounding Map Refinement (SMR) module to obtain fine sounding maps containing shape information. SMR predicts semantic affinities of paired adjacent coordinates in the image by inter-pixel relation mining. Then, SMR produces fine sounding maps by revising the coarse sounding maps according to the affinity matrix. After these processes, SMR significantly refines the coarse sounding maps, enabling object boundary information to be recovered.

Although SMR produces fine sounding maps, it is too heavy for evaluation. Thus we propose Sounding Pixel Segmentation (SPS) module. SPS is an encoder-decoder architecture that includes an audio signal supervised. We use the fine sounding maps generated by SMR to train the SPS. We also find that SPS can improve the training of SMR to obtain better convergence. Therefore, SPS makes our methods more stable and accurate in sounding map generation.

Our major contributions are summarized below:

- We design a novel mask augmentation based multi-instance contrastive learning that alleviates the overfit and sub-optimization problem to realize audio-visual alignment better.

- We propose an unsupervised sounding map refinement (SMR) module that leverages the visual semantic affinity to recover the sounding object boundary and produce fine-grained sounding maps.

- We adopt a sounding pixel segmentation (SPS) module for lightweight evaluation and facilitating the SMR converge to better performance.

- The final segmentation results outperform SOTA unsupervised methods by 9.25% in mIoU on the AVSBench-S4 and by 2.62% in cIoU on the VGGSS benchmark.

## 2 Related Work

### 2.1 Sounding source localization

Sound Source Localization (SSL) aims to locate the sounding regions in visual frames.(Wei et al., 2022) Recent approaches (Arandjelovic and Zisserman, 2018; Senocak et al., 2018; Hu et al., 2019; Afouras et al., 2020; Hu et al., 2020; Liu et al., 2022a; Zhou et al., 2022; Song et al., 2022; Liu et al., 2022b; Sun et al., 2023) extensively leveraged contrastive learning based on audio-visual co-occurrence to address this issue unsupervised as shown in Fig.1(a). For example, (Arandjelovic and Zisserman, 2018) formulated the problem in a Multiple Instance Learning (MIL) framework and trained a localization network in the manner of Audio-Visual Correspondence(Arandjelovic and Zisserman, 2017). Accordingly, (Chen et al., 2021) incorporated explicitly background regions with low correlation to the given audio into the framework and regarded them as hard negative in contrastive learning. (Sun et al., 2023) investigated the false negative issue in audio-visual contrastive learning and proposed a method to alleviate this problem. In (Qian et al., 2020), the authors explored for multi-source localization.

However, these unsupervised SSL methods only produce inaccurate patch-level results and our goal is to produce fine-grained pixel-level sounding results. The work most closely related to ours is AVS (Zhou et al., 2022), which employed a segmentation network trained with label supervision to produce pixel-level sounding maps, as shown in Figure 1(b). However, this work is difficult to apply to real scenarios, because it needs burden-intensive human annotations. Therefore, we focus on unsupervised pixel SSL which aims to produce pixel-level localization results for sounding objects in videos without any annotations.

### 2.2 Weakly-supervised semantic segmentation

The approaches of weakly supervised semantic segmentation usually start training a classification

network with image-level labels and produce the initial pseudo labels using CAM. To address the drawback of incomplete object activation of CAM, these methods always build a refined process. (Ahn and Kwak, 2018) involved modeling the pixel-level affinity distance from initial CAMs and employed a random walk to propagate individual class labels at the pixel level to refine the result. The random walk(Vernaza and Chandraker, 2017) is an operation propagating the sparse labels to produce guessed dense labels which are often used in applying affinity matrix to refine map. (Ru et al., 2022) built a transformer encoder and learned the semantic affinities from the multi-head self-attention. Then it leveraged the learned affinity to refine the initial pseudo labels for segmentation. These methods usually adopt pixel-level refinement algorithms *e.g.* dCRF(Krähenbühl and Koltun, 2011), PRM(Araslanov and Roth, 2020), PAMR (Ru et al., 2022) to help refining. In this work, we build a Sounding Map Refinement(SMR) module to refine the coarse sounding map which is similar to (Ahn and Kwak, 2018).

## 3 Methodology

In this section, we introduce the proposed method **Unsupervised Sounding Pixel Learning** (USPL) to conduct the pixel SSL task. In USPL, we first align the audio and visual features in Section 3.1 to obtain the coarse sounding maps. Then, we propose a refined module named Sounding Map Refinement (SMR) to recover the boundary from the coarse sounding maps in Section 3.2. Next, we apply the fine sounding maps generated by SMR to train the Sounding Pixel Segmentation (SPS) module in Section 3.3.

### 3.1 Unsupervised cross-modality coarse localization

Given an audio-visual dataset $D = \{(v_i, a_i) : i = 1, ..., N\}$, we extract two modality features by encoders $f_a(\cdot)$, $f_v(\cdot)$, respectively. The audio encoder, here we use a pretrained VGGish, extracts global audio features $F_a^i = f_a(a_i), F_a^i \in \mathbb{R}^C$. The visual encoder, here we use a pretrained ResNet50, produces a set of features spanning all locations in an image as $F_v^i = f_v(v_i), F_v^i \in \mathbb{R}^{h \times w \times c}$. Then, we utilize two projectors $g_a(\cdot)$ and $g_v(\cdot)$, one for each modality feature, to map the features into a shared feature space. The projection process can be realized by $\hat{F}_a^i = g_a(F_a^i)$, $\hat{F}_v^i = g_v(F_v^i)$, where the $\hat{F}_a^i$

and $\hat{F}_v^i$ are features after projectors mapping. We compute the audio-visual cross attention to obtain the coarse sounding maps $M_{\text{crs}}$ in Eq. 1.

$$M_{\text{crs}} = \frac{\langle F_a, F_v \rangle}{\|F_a\| \cdot \|F_v\|} \qquad (1)$$

**Mask augmentation based Multi-Instance Contrastive Learning** (MMICL) Following the previous EZ-VSL (Mo and Morgado, 2022b), we conduct multiple-instance contrastive learning (MICL) to align the audio-visual features. Specifically, we define the visual features from various locations in the paired audio-visual set as positive bag $P_v^i = \{\hat{F}_v^i\}$ while those from unpaired sets as negative bags $N_v^i = \{\hat{F}_v^t, t = 1...N, t \neq i\}$. Obviously, the audio feature is matched with sounding positions visual feature in the positive bag $P_v^i$ while mismatched at all visual features from negative bags $N_v^i$.

Following the mentioned MICL, we find that the model is prone to overfit and often locates part of sounding objects. Most likely because the MICL makes an inappropriate assumption that the audio feature is matched at only one visual location in the positive bag. This assumption makes the model easy to overfit in focusing on the most matched visual regions and failing to locate all sounding positions. To combat overfitting and mine the complete sounding objects, we propose Mask augmentation based Multi-Instance Contrastive Learning(MMICL) when training. It contains a random mask augmentation layer $\delta$ which randomly drops the visual features of some locations during training to ensure potential sounding locations have more opportunities for optimization. The MMICL loss is defined in Eq. 2.

$$\mathcal{L}_{mmicl} = -\sum_i log \frac{exp(\frac{1}{\tau} sim(\hat{F}_a^i, \delta(\hat{F}_v^i)))}{\sum_j exp(\frac{1}{\tau} sim(\hat{F}_a^i, \hat{F}_v^j))} \qquad (2)$$

where $\tau$ is the temperature parameter, and the similarity $sim(\hat{F}_a^i, \hat{F}_v^i)$ between an audio feature $\hat{F}_a^i$ and a bag of visual features $\hat{F}_v^i$ is computed by max-pooling audio-visual cross attention.

### 3.2 Unsupervised Sounding Map Refinement

In order to localize sounding objects with high precision, we build a Sounding Map Refinement (SMR) module to recover the object shape information from the coarse sounding maps $M_{\text{crs}}$.

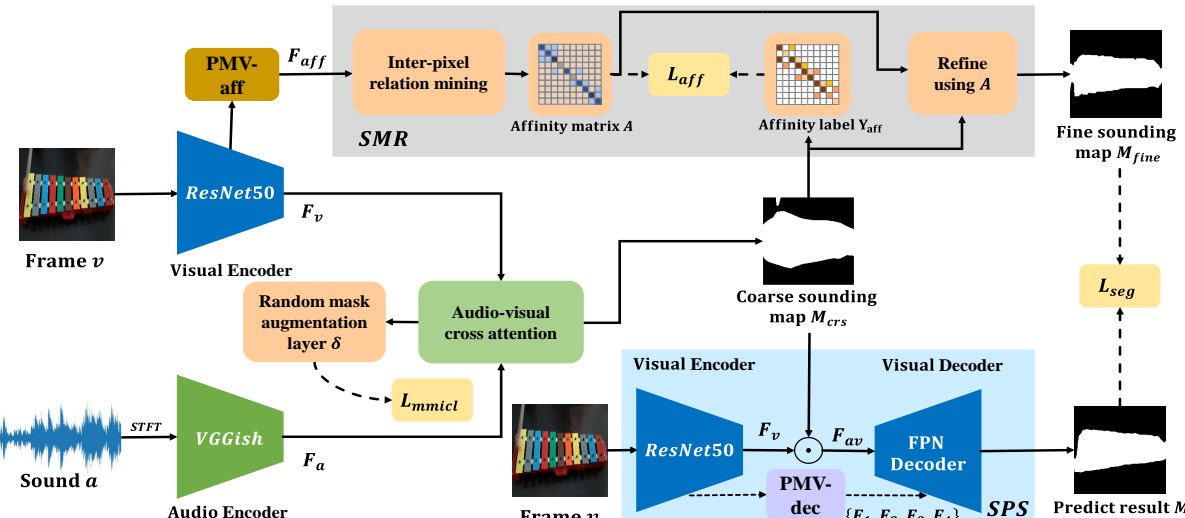

Figure 2: **The proposed Unsupervised Sounding Pixel Learning (USPL) method.** We first extract paired audio-visual feature $F_a$, $F_v$ using each modality extractor. Then, we propose a mask augmentation based multi-instance contrastive learning which aligns the audio-visual feature to generate coarse sounding maps $M_{crs}$. In the Sounding Map Refinement (**SMR**) module, we derive the affinity matrix $A$ from the affinity feature $F_{aff}$ by mining the inter-pixel relation in the image. Next, we employ the learned affinity to revise the $M_{crs}$ and output fine sounding maps. The Sounding Pixel Segmentation (**SPS**) module produces the final segmentation sounding map results. The Process of Multi-scale Visual feature (**PMV**) block contains two process streams to provide a multi-scale feature for SMR and SPS which is detailed in Fig.3

Specifically, SMR builds an affinity matrix to predict the semantic affinities between adjacent coordinate features in an image. This involves analyzing the relationship between different locations of image features. Once affinity matrix $A$ has been predicted, SMR uses $A$ as transition probabilities to revise coarse sounding map.

**Semantic affinity.** As shown in Fig.2, SMR first computes the affinity matrix $A$ from the affinity feature $F_{aff}$. The affinity feature is produced by the PMV-aff block which is detailed in Fig. 3. To generate the affinity feature $F_{aff}$, we concatenate the multi-scale visual features extracted from ResNet50 and then apply an MLP layer to aggregate these hierarchical features. After extracting the affinity feature, we compute the Affinity matrix $A$ by inter-pixel relation mining. Specifically, we identify a set of locations that are within a distance of $r$ in coordinate space. The semantic affinity between neighborhood features $i$ and $j$ in this set is denoted by $A(i, j)$ defined in Eq. 3.

$$A(i, j) = exp\left\{-\left\|F_{aff}(x_i, y_i) - F_{aff}(x_j, y_j)\right\|_1\right\} \quad (3)$$

The SMR module needs semantic affinity labels for paired features to confirm their relations. However, because of the unlabeled setting, no affinity labels can be obtained directly. In such case, we generate affinity labels from the coarse sounding map $M_{crs}$ instead. In the following paragraphs, we present

how to generate affinity labels and how to train the SMR with affinity labels.

For the purpose of training SMR to generate the affinity matrix $A$, we first obtain semantic affinity labels $Y_{aff} \in \mathbb{R}^{hw \times hw}$ from the coarse sounding map $M_{crs}$. Our approach involves identifying confident areas of the sounding object and background from the coarse sounding map and then sampling training examples from these regions. In order to obtain better alignment with the low-level image appearance, we use a pixel-level refinement algorithm Pixel-wise Adaptive Mean Shift (PAMR) (Ru et al., 2022) to refine the coarse sounding maps in Eq. 4.

$$M_{rf} = \text{PAMR}\left(M_{crs}\right) \quad (4)$$

Then, we employ two background thresholds $\beta_l$ and $\beta_h$, where $0 < \beta_l < \beta_h < 1$, to generate a Tri-map $Y_{tri}$ which delineates the reliable sounding area, silent area, and uncertain area as defined in Eq. 5.

$$Y_{tri}(x, y) = \begin{cases} 1, & \text{if } M_{rf} \geq \beta_h \\ 0, & \text{if } M_{rf} < \beta_l \quad (5) \\ 255, & \text{otherwise} \end{cases}$$

where 1 and 0 denote the sounding area and the silent area, and 255 is the ignore area. $Y_{tri}(x, y)$ is the tri-map of coordinate $(x, y)$.

The pseudo affinity label $Y_{aff}$ is then derived from $Y_{tri}$. Specifically, in tri-map $Y_{tri}$, if the $(x_i, y_i)$

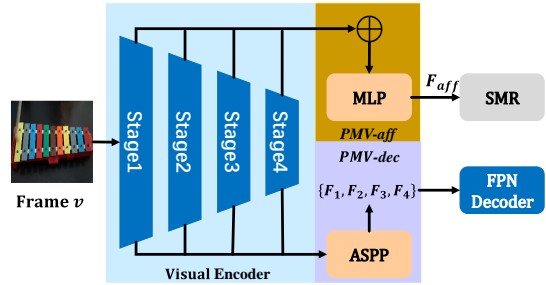

Figure 3: The Process of Multi-scale Visual feature (**PMV**). To leverage the multi-scale visual features in SMR and SPS, we extract multi-stage features from ResNet50. Subsequently, two post-processing blocks, **PMV-aff** and **PMV-dec**, are constructed to produce the affinity feature $F_{\mathrm{aff}}$ and a residual feature set $\{F_1, F_2, F_3, F_4\}$, which are then passed to subsequent modules. $\oplus$ means concatenate.

and $(x_j, y_j)$ are from the same semantic area, we set their affinity labels as positive $Y_{\mathrm{aff}}^+$; otherwise, their affinity labels are set as negative $Y_{\mathrm{aff}}^-$. If either $(x_i, y_i)$ or $(x_j, y_j)$ is sampled from the ignored area, their affinity labels will be set as ignored. Once the affinity label $Y_{\mathrm{aff}}$ has been obtained, we can train the SMR under supervision to predict the semantic affinity matrix $A$ as defined in Eq. 6.

$$\mathcal{L}_{\mathrm{aff}} = -Y_{\mathrm{aff}}^+ \cdot log(A) - Y_{\mathrm{aff}}^- \cdot log(1 - A) \quad (6)$$

where $Y_{\mathrm{aff}}^+$ is the positive label in $Y_{\mathrm{aff}}$ and $Y_{\mathrm{aff}}^-$ is the negative one.

**Sounding map refinement.** The learned reliable semantic affinities between paired adjacent coordinate features could be utilized to refine the coarse sounding maps. This process is achieved through random walk, as demonstrated in (Vernaza and Chandraker, 2017). For the learned semantic affinity matrix $A$, the semantic transition matrix $T$ is defined by Eq. 7.

$$T = D^{-1}A^\beta, where D_{ii} = \sum_j A_{ij}^\beta \quad (7)$$

where $\beta$ is a hyper-parameter to ignore trivial affinity values in $A$, and $D$ is a diagonal matrix to normalize $A$. The propagation of refinement for the coarse sounding map $M_{\mathrm{crs}}$ is defined by Eq. 8.

$$M_{\mathrm{fine}} = T^t \times vec(M_{\mathrm{crs}}) \quad (8)$$

where $vec(\cdot)$ means vectorization of a matrix, $M_{\mathrm{fine}}$ is the fine sounding map. This propagation process diffuses the semantic regions with high

affinity and suppresses the wrongly activated regions so that the activated maps align better with the semantic boundaries.

After the refinement, the fine sounding maps $M_{\mathrm{fine}}$ are fine-grained enough for training sounding pixel segmentation networks.

### 3.3 Sounding Pixel Segmentation (SPS)

Although the SMR produces fine-grained sounding maps, it is too heavy for evaluation. Thus, we adopt the Sounding Pixel Segmentation (SPS) module as shown in Fig.2. We also observe that SPS can improve the training performance of SMR to produce more accurate localization results.

**Audio-visual interactive Encoder.** As illustrated in Fig.2, the visual encoder of SPS is structurally equivalent to the encoder employed in MMICL, and these two encoders share weights. Therefore, we use the visual feature extracted in Sec. 3.1 for SPS. Here we use the coarse sounding map $M_{crs}$ to fuse the audio-visual features as defined in Eq. 9.

$$F_{av} = F_v + F_v \odot norm(M_{crs}) \quad (9)$$

where $F_{av}$ is the audio visual fusion feature, $norm$ is Min-Max Normalization. $\odot$ is Hadamard product.

As shown in Fig.3, after getting multi-scale visual features from various stages, we use Atrous Spatial Pyramid Pooling (ASPP) (Chen et al., 2017) to produce a residual feature set $\{F_1, F_2, F_3, F_4\}$ for subsequent processing.

**Segmentation decoder.** We here use the FPN-decoder(Kirillov et al., 2019) in this work for its flexibility and effectiveness. The audio-visual fusion feature and residual feature set are fed into the decoder. In short, at the $j$-th decoder layer, both the feature from residual feature set $F_{5-j}$ and decoder stage $F_{j-1}^d$ are utilized for the decoding process. The decoded features are then upsampled to the next stage. The final output of the decoder is $M \in \mathcal{R}^{T \times H \times W}$, activated by $sigmoid$.

Given the prediction $M$ and the SMR fine sounding map $M_{\mathrm{fine}}$, we straightforwardly utilize the binary cross-entropy loss, similar to supervised training for network training which is defined in Eq. 10.

$$\mathcal{L}_{seg} = -M_{\mathrm{fine}} \cdot log(M) \quad (10)$$

### 3.4 Loss settings

As shown in Fig.2, our framework comprises three loss terms, a mask augmentation based multi-

| Methods | Sup. | Modality | mIoU (%) | F-score |
|---|---|---|---|---|
| 3DC(Mahadevan et al., 2020) | $\mathcal{F}$ | V | 57.10 | 0.759 |
| iGAN(Mao et al., 2021) | $\mathcal{F}$ | V | 61.59 | 0.801 |
| AVS (Zhou et al., 2022) | $\mathcal{F}$ | A+V | 78.74 | 0.879 |
| LAVISH(Lin et al., 2023) | $\mathcal{F}$ | A+V | **80.10** | **0.891** |
| CAM(Zhou et al., 2016) | $\mathcal{I}$ | V | 20.40 | 0.387 |
| IRNet(Ahn et al., 2019) | $\mathcal{I}$ | V | 35.68 | 0.475 |
| MSSL(Qian et al., 2020) | $\mathcal{I}$ | A+V | **44.89** | **0.663** |
| FNAC(Sun et al., 2023) | $\mathcal{U}$ | A+V | 32.46 | 0.434 |
| EZ-VSL(Mo and Morgado, 2022b) | $\mathcal{U}$ | A+V | 34.01 | 0.562 |
| LVS (Chen et al., 2021) | $\mathcal{U}$ | A+V | 37.94 | 0.510 |
| USPL(Ours) | $\mathcal{U}$ | A+V | **47.19** | **0.617** |

Table 1: Performance on AVSBench-S4 test, compared to several related methods. *Sup.* denotes supervision types. $\mathcal{F}$: train under full supervision; $\mathcal{I}$: train under weakly supervision with image-level label; $\mathcal{U}$: train under unsupervision with no label. *Modality* denotes the modalities of input data. V refers to visual only; A+V refers to audio and visual.

instance contrastive learning loss $\mathcal{L}_{mmicl}$, a refine training loss $\mathcal{L}_{aff}$, and a segmentation loss $\mathcal{L}_{seg}$. Since no annotations exist in our method, the model heavily relies on the information obtained from correspondence audio-visual data, so we first only use $\mathcal{L}_{mmicl}$ training the model for 3 epochs to make our model more stably. After that, we use the overall loss defined in Eq.11 to train the model. The overall loss is weighted sum of $\mathcal{L}_{mmicl}$, $\mathcal{L}_{aff}$, $\mathcal{L}_{seg}$.

$$\mathcal{L} = \lambda_1 \mathcal{L}_{mmicl} + \lambda_2 \mathcal{L}_{aff} + \lambda_3 \mathcal{L}_{seg} \qquad (11)$$

where $\lambda_1$, $\lambda_2$, and $\lambda_3$ serve as weighting factors that balance the contributions of various loss functions.

## 4 Experiments

### 4.1 Dateset and Implementation Details

**Dataset.** We evaluate our approach on two datasets: AVSBench-S4(Zhou et al., 2022) and VGGSound (Chen et al., 2020a). AVSBench-S4 is the single source subset of the AVSBench dataset which contains 4,932 videos of $5s$ over 23 categories. VGGSound dataset is a large video dataset containing over 200K videos from 309 categories.

Localization performances are measured on two benchmarks, AVSBench-S4 test and VGGSound Source (VGGSS)(Chen et al., 2021). The AVSBench-S4 test has 3700 audio-visual pairs with manually pixel-level sounding object labels. VGGSS has 5,000 audio-visual pairs with manual boundary box-level sounding object labels.

**Implementation Details.** Our visual inputs are the first frame of each clip in AVSBench-S4 and the center frame of videos in VGGSound.

The images are resized to $224 \times 224$ resolution. The audio inputs of the AVSBench-S4 dataset are $1s$ clips and the audio inputs of VGGSound are $3s$ clips that contain the selected frame. Here the raw audio waveform has been converted to a $257 \times 301$ log-mel spectrogram by Short-Time Fourier Transform (STFT)(Aytar et al., 2016). We adopt a VGGish(Hershey et al., 2017) pretrained on Audio-Set as audio encoder and a slightly modified ResNet50(He et al., 2016) pretrained on ImageNet as visual encoder, respectively. We modify the last conv-stride of Resnet50. Besides, we remove the last linear layer and pooling layer of Resnet50.

As shown in Fig.3, the dim of affinity feature extracted by PMV-aff is 256, the dim of multi-scale residual feature set $[F_1, F_2, F_3, F_4]$ are $[256, 512, 1024, 2048]$. The dropout probability of random mask augmentation layer $\delta$ is set to 0.5. We set the cared neighborhood radius $r = 7$. We set the background scores $\beta_l = 0.4$ and $\beta_h = 0.55$ in Eq. 5, respectively. The resolution of affinity matrix $A$ in our experiment is $56 \times 56$. We set $\beta = 10$ in Eq.7 and $t = 8$ in Eq.8. The temperature parameter $\tau = 0.07$. The balanced parameters in overall loss of Eq.11 are $\lambda_1 = 0.1$, $\lambda_2 = 0.4$, $\lambda_3 = 0.5$. The batch size is set as 32. All models are trained with the Adam optimizer using a learning rate of $10^{-4}$. For training stability, we warm up by MMICL loss for 3 epochs, then we train the model using the overall loss for the remaining 20 epochs.

### 4.2 Comparison with SOTAs

We use two common metrics of semantic segmentation, Mean Intersection over Union (mIoU) and

| Methods | *Sup.* | *Modality* | cIoU-0.5 (%) | AUC (%) |
|---|---|---|---|---|
| Attention10k(Senocak et al., 2018) | $\mathcal{U}$ | A+V | 18.50 | 30.20 |
| DMC(Hu et al., 2019) | $\mathcal{U}$ | A+V | 29.10 | 34.80 |
| AVObject(Afouras et al., 2020) | $\mathcal{U}$ | A+V | 29.70 | 35.70 |
| LVS(Chen et al., 2021) | $\mathcal{U}$ | A+V | 34.40 | 38.20 |
| SSPL(Song et al., 2022) | $\mathcal{U}$ | A+V | 33.90 | 38.00 |
| EZ-VSL(Mo and Morgado, 2022b) | $\mathcal{U}$ | A+V | 34.38 | 37.70 |
| EZ-VSL(Mo and Morgado, 2022b) | $\mathcal{U} + \mathcal{O}$ | A+V | 38.85 | 39.54 |
| SLAVC(Mo and Morgado, 2022a) | $\mathcal{U}$ | A+V | 37.22 | 38.60 |
| SLAVC(Mo and Morgado, 2022a) | $\mathcal{U} + \mathcal{O}$ | A+V | 39.67 | 39.11 |
| FNAC(Sun et al., 2023) | $\mathcal{U}$ | A+V | 39.50 | 39.66 |
| FNAC(Sun et al., 2023) | $\mathcal{U} + \mathcal{O}$ | A+V | 41.85 | 40.80 |
| USPL(Ours) | $\mathcal{U}$ | A+V | **44.47** | **45.11** |

Table 2: Performance on VGGSS dataset, compared to various SSL methods. *Sup.* denotes supervision types. $\mathcal{U}$: train under unsupervised with no label; $\mathcal{U} + \mathcal{O}$: unsupervised training but evaluate with the help of a trick OGL(Mo and Morgado, 2022b); *Modality* denotes the modalities of input data. A+V refers to audio and visual.

F-score to evaluate in the AVSBench-S4 test. Additionally, we use another two metrics to evaluate in VGGSS, Consensus Intersection over Union(cIoU) (Senocak et al., 2018) and Area Under Curve (AUC). For all metrics, high values mean better localization performances.

**AVSBench-S4.** As shown in Table 1, we present the comparisons between various approaches on the AVSBench-S4. Our proposed method USPL gets 47.19% mIoU and 0.617 F-score in the AVSBench-S4. It shows that our USPL clearly outperforms previous SOTA unsupervised methods LVS by 9.25% mIoU and 0.11 F-score respectively, while even competing with some weakly supervised methods, such as CAM, and MSSL. It is worth mentioning that our unsupervised method achieves a remarkable mIoU score of 47.19%, which reaches 58.91% of the best-supervised counterpart LAVISH(Lin et al., 2023) on AVSBench-S4 test.

**VGGSS.** As shown in Table 2. Our method also beats all previous SSL methods with a significant improvement. For example, on the VGGSS benchmark, we outperform previous SOTA FNAC by 2.62% cIoU and 4.31% AUC. It's noticed that some methods use object-guided localization (OGL)(Mo and Morgado, 2022b) trick to improve the performance. This trick introduces the visual prior knowledge from the pretrained dataset to help localization. We consider that the OGL violates the assumption of unsupervision. Therefore, we don't adopt the OGL in our method and we still get the best performance.

### 4.3 Ablation Study and Analysis

**Module ablation.** Our USPL contains three main modules as shown in Fig.2. To explore the impact of each module, we ablate each component individually in Table 3. All results are obtained following the same parameter setting on AVSBench-S4 val and test set. The first row is our baseline EZ-VSL, which gets 33.85% mIoU on the validation set and 34.01% mIoU on the test set. The second row is the result of using augmentation based multi-instance contrastive learning which achieves a 0.24% promotion on the validation set than baseline. The third row is the result of adding SMR block which boosts performance by visual affinity refinement, achieving a 9.63% mIoU promotion on validation set. The fourth row is the result of ablating SMR block while using MMICL and SPS block, which drops 10.08% mIoU comparing with adding SMR. The last row is the result of adding the SPS block which gets a more accurate result, achieving 45.41% mIoU on the validation set.

As discussed in Section 3.3, the proposed SPS module can help the convergence of SMR to reach batter performance. We here conduct an experiment to prove this. As shown in Table 4, when training SMR alone, the mIoU of the fine sounding map produced by SMR in AVSBench-S4 test is 44.01%, when we train the SMR together with SPS, the mIoU of the fine sounding map produced by SMR has increased to 46.07%.

**Analysis of module efficiency.** Here we discuss the model computation and memory costs as shown in Table 3. In our proposed USPL, SMR

| | modules | | | mIoU(%)↑ | | results | | |
|---|---|---|---|---|---|---|---|---|
| | MMICL | SMR | SPS | val | test | Params(M)↓ | FLOPS(G)↓ | FPS↑ |
| EZ-VSL (Baseline) | | | | 33.85 | 34.01 | 24.14 | 6.46 | 15.45 |
| USPL (Ours) | √ | | | 34.09 | 35.52 | 24.14 | 6.46 | 15.45 |
| | √ | √ | | 43.72 | 44.01 | 69.64 | 40.85 | 4.34 |
| | √ | | √ | 35.49 | 35.61 | 68.12 | 38.61 | 9.94 |
| | √ | √ | √ | **45.41** | **47.19** | 68.12 | 38.61 | 9.94 |

Table 3: Ablation study of MMICL, SMR, and SPS module of the proposed USPL evaluated on the val and test set of AVSBench-S4 in mIoU(%). Analysis computation and memory cost of baseline and USPL's modules in inference procedure.

| | mIoU% |
|---|---|
| Only SMR | 44.01 |
| SMR & SPS | 46.07 |

Table 4: Ablation study of SMR module. We investigate the effects of the SPS module on the SMR module.

and SPS modules play crucial roles. SMR module refines the sounding maps by leveraging visual semantic affinity, resulting in fine-grained sounding maps. However, it is noted that SMR exhibits low efficiency. Therefore, we propose SPS module serving as a lightweight module for evaluation. As shown in Table 3, we calculate the parameters (Params), floating point operations (FLOPS), and frames per second (FPS) to evaluate the methods' efficiency. We find that using SPS to bypass SMR could not only reduce space complexity and significantly decrease time complexity, but also lead to better performance. However, compared to the baseline, our USPL exhibits lower efficiency both in time and space complexity. We have noticed this problem and we will improve it in future work.

| $\delta$ | mIoU(%) | F-score |
|---|---|---|
| 0 | 43.07 | 0.545 |
| 0.1 | 44.87 | 0.564 |
| 0.3 | 45.12 | 0.561 |
| **0.5** | **45.41** | **0.565** |
| 0.7 | 44.63 | 0.554 |
| 0.9 | 13.49 | 0.223 |

Table 5: Evaluation on the drop probability $\delta$ of Random mask augmentation layer on AVSBench-S4 validation set.

**Hyper-parameters.** In Table Table 5, we analyze the hyper-parameters $\delta$ in MMICL module. The drop probability $\delta$ in the random mask augmentation layer controls the probability of visual features dropping during training. As shown in Table 5, we perform a series of settings and compare the results to the case where no random mask augmentation is used, i.e., $\delta = 0$. It is observed that using a random mask always yields better performance. Moreover, when $\delta = 0.5$, the model achieves its optimal performance. It is also important to note that excessive drop probability can be detrimental to the model's performance.

| radius $r$ | mIoU(%) | F-score |
|---|---|---|
| 3 | 42.27 | 0.531 |
| 5 | 43.77 | 0.551 |
| 7 | **45.41** | **0.565** |
| 9 | 39.59 | 0.502 |

Table 6: Evaluation on the radius $r$ of affinity matrix generation on AVSBench-S4 validation set.

Subsequently, we conduct ablation studies on the hyper-parameter in the SMR module as shown in Table 6. The parameter $r$ controls the radius of the affinity matrix generation, which determines the scope of interest neighborhood around each point in the coarse sounding maps. In our experiments, we find that setting $r = 7$ yields the best results. It's noted that excessive values of $r$ may generate unnecessary relations that can adversely affect the final outcome. Next, we explore the hyper-parameter in Eq. 5, specifically focusing on the two background thresholds $\beta_l$ and $\beta_h$ which control the Tri-map division. As shown in Table 7, we establish that setting $\beta_l$ to 0.4 and $\beta_h$ to 0.55 yields optimal results, respectively. Finally, as shown in Table 8, we analyze the hyper-parameters of Eq. 7 and Eq. 8, which significantly impact sounding map refinement. The $\beta$ in Eq. 7 is for ignoring trivial affinity values in the semantic affinity matrix, while the $t$ in Eq. 8 controls the iterations of the semantic transition matrix. Our parameters search

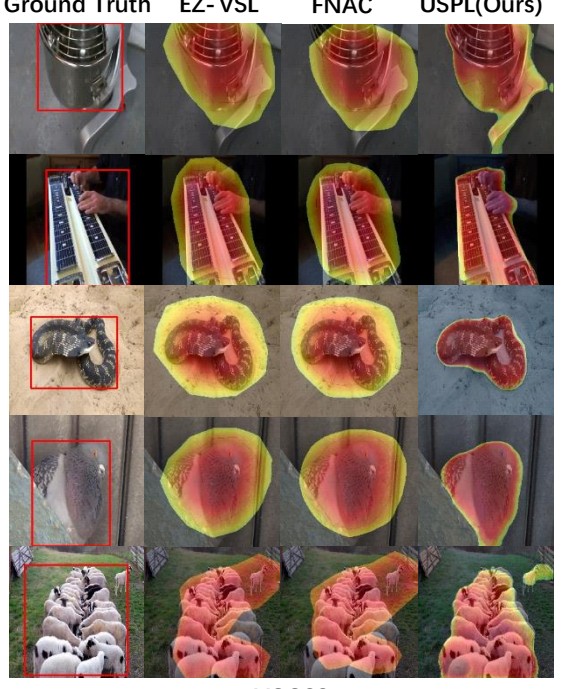 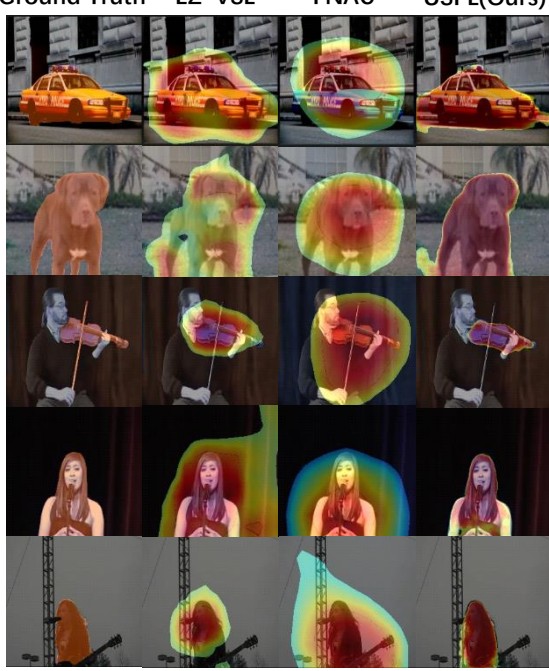

VGGSS           AVSBench - S4

Figure 4: **Qualitative results** for USPL training on AVSBench-S4 (right) and VGGSS (left). For each dataset, the first column shows annotations overlaid on images, and the following two column show predictions trained on Unsup SSL methods, EZ-VSL (Mo and Morgado, 2022b) and FNAC(Sun et al., 2023). The last column shows our USPL prediction results. Obviously, USPL produces more accurate sounding pixel prediction results than the previous SSL method.

through grid-based evaluation reveals that setting $\beta$ = 10 and $t$ = 8 yields optimal results for our model.

| $\beta_h$ | $\beta_l$ | mIoU(%) | F-score |
|-----------|-----------|---------|---------|
| 0.75 | 0.6 | 30.69 | 0.473 |
| 0.65 | 0.5 | 37.94 | 0.527 |
| **0.55** | **0.4** | **45.41** | **0.565** |
| 0.45 | 0.3 | 41.19 | 0.492 |

Table 7: Evaluation on the background thresholds $\beta_h$, $\beta_l$ of Tri-map generation on AVSBench-S4 validation set.

| mIoU(%) | | $t$ | | | |
|---------|---|-----|---|---|---|
| | | 1 | 4 | **8** | 12 |
| | 1 | 39.53 | 34.26 | 15.19 | 15.08 |
| | 5 | 40.49 | 43.68 | 36.61 | 13.18 |
| $\beta$ | **10** | 40.07 | 42.51 | **45.41** | 24.03 |
| | 15 | 39.75 | 41.65 | 44.95 | 37.77 |

Table 8: Evaluation on the hyper-parameters $\beta$, $t$ of the sounding map refinement on AVSBench-S4 validation set, reported in mIoU.

### 4.4 Qualitative Results

As shown in Fig. 4, we show some qualitative comparisons between previous Unsup SSL methods, EZ-VSL, FNAC, and our USPL on the AVSBench-S4 test and VGGSS benchmark. As we can observe, compared to Unsup SSL methods, our method generally produces more accurate localization results. As a pixel-level method, the predictions of USPL tend to be complete and highly consistent with the shape of the sounding objects, which means, a precise prediction of the object boundaries, while EZ-VSL and FNAC often locate part of the sounding

object or predict an approximate region of sounding object.

### 5 Conclusion

In this work, we propose USPL, a novel unsupervised approach for pixel SSL. Specifically, we first use the MMICL module to align audio-visual features and obtain coarse sounding maps. Then, we propose an SMR module to refine the coarse sounding maps by the visual semantic affinity. Next, we propose SPS to produce final segmentation results. Our approach shows promising performance on pixel SSL tasks, especially achieving new SOTAs on the AVSBench-S4 test and VGGSS benchmark.

## Limitations

Although USPL behaves well on pixel SSL at single sound source localization, it is not applicable to localize multiple sound sources in unconstrained videos (Qian et al., 2020), which is still a challenge for the community. A potential solution is to develop weakly- or semi-supervised methods. We leave it for future work. Another limitation that we need to address in our work is the effective integration of temporal information. As video data is inherently a time series, it contains a wealth of information that is embedded within the temporal dimension. By solely utilizing a single frame, as done in previous SSL methods, we risk losing valuable temporal context. Therefore, in our future work, we will prioritize incorporating temporal information into our approach.

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
