# OpenReview forum: "Unsupervised Sounding Pixel Learning"
_EMNLP/2023/Conference — EMNLP 2023 Main_

### Official Review · Reviewer_fozg · 2023-08-01

**Soundness:** 4

**Excitement:**

4: Strong: This paper deepens the understanding of some phenomenon or lowers the barriers to an existing research direction.

**Paper Topic And Main Contributions:**

The paper addresses the problem of unsupervised sound source localization in videos. The authors propose an approach called Unsupervised Sounding Pixel Learning (USPL) that aims to accurately localize sounding objects in a given audio-visual scene without any annotations. They propose three main modules: (1) Mask augmentation based multi-instance contrastive learning to obtain coarse sounding maps, (2) Unsupervised Sounding Map Refinement (SMR) module to refine the coarse maps and recover object boundary information, and (3) Sounding Pixel Segmentation (SPS) module to produce the final pixel-level segmentation results.

The main contributions of the paper are as follows:

1. The authors propose a novel approach, USPL, for unsupervised pixel-level sound source localization in videos. They address the limitation of previous unsupervised methods that focus on coarse localization by aiming to produce fine-grained pixel-level localization results.

2. The proposed method includes a mask augmentation based multi-instance contrastive learning module, which alleviates the problem of overfitting and sub-optimization in previous SSL methods. This module aligns audio-visual features to obtain coarse sounding maps.

3. The authors introduce the Unsupervised SMR module that leverages visual semantic affinity to recover the sounding object boundary and produces fine sounding maps. This module contributes to the recovery of object boundary information from the coarse maps obtained in the previous step.

4. The SPS module is presented to realize audio-supervised semantic segmentation. This module improves the training performance of SMR to predict more accurate pixel-level sounding localization results.

The authors conduct extensive experiments on the AVSBench-S4 and VGGSound datasets and show improved performance compared to previous state-of-the-art unsupervised methods. The results demonstrate the superiority of the proposed approach in terms of mIoU and cIoU metrics.

**Questions For The Authors:**

A. Have you attempted to leverage the temporal information between frames in your analysis? Could this approach provide any benefits?

B. Have you assessed the contribution of audio to the model? How does the performance vary when audio information is not provided? Can the model still achieve satisfactory performance by merely predicting the mask of the salient object in the image?

**Reasons To Accept:**

Strengths:

1. Novel approach: The paper proposes an unsupervised method USPL for pixel-level sound source localization. It addresses the challenges of learning accurate pixel-level sounding maps and fine-grained sound source localization in an unsupervised paradigm.

2. Multi-stage approach: The paper includes three main modules - mask-augmentation based multi-instance contrastive learning, unsupervised sounding map refinement, and sounding pixel segmentation. This multi-stage approach allows for the alignment of audio-visual features, refinement of coarse sounding maps, and generation of fine sounding maps.

3. Experimental results: The proposed USPL method outperforms previous state-of-the-art unsupervised methods in terms of mIoU and F-score. It achieves a significant improvement in sound source localization accuracy on both the AVSBench-S4 and VGGSound datasets.

Benefits to the community:

1. Unsupervised sound source localization: The USPL method presents a valuable contribution to the community by addressing the challenge of unsupervised sound source localization. By developing an approach that does not rely on manual annotations, it provides a more efficient and scalable solution for this task.

2. Accurate pixel-level localization: The USPL method achieves accurate pixel-level sound source localization, which is crucial for various applications, such as audio event detection, speech recognition, and audio-visual scene understanding. It provides researchers and practitioners with a reliable tool for extracting precise information from audio-visual data.

3. Superior performance: The experimental results demonstrate the superiority of the proposed USPL method compared to existing unsupervised methods. The improved localization accuracy can benefit researchers in developing more accurate and robust models for various audio-related tasks.

**Reasons To Reject:**

While the paper presents a detailed description of the proposed method, it lacks a deeper theoretical analysis and explanation of the underlying principles supporting the effectiveness of the approach.

**Reproducibility:**

3: Could reproduce the results with some difficulty. The settings of parameters are underspecified or subjectively determined; the training/evaluation data are not widely available.

**Reviewer Confidence:**

4: Quite sure. I tried to check the important points carefully. It's unlikely, though conceivable, that I missed something that should affect my ratings.

---

> ### Author Rebuttal · Authors · 2023-08-28
>
> Thank you for your comprehensive review of our paper. Your attention to detail and expertise have contributed significantly to the improvement of our work. We are thankful for your support and look forward to incorporating your suggestions into our future work.
>
> **Q1** : While the paper presents a detailed description of the proposed method, it lacks a deeper theoretical analysis and explanation of the underlying principles supporting the effectiveness of the approach.
>
> **Answers :** Thank you very much for valuable suggestion.
>
> We will add deeper theoretical analysis and explanation supporting to our method.
>
> Our method is based on two facts. One is that huge numbers of videos exist, and it is hard to apply traditional supervised segmentation methods on these huge video data because of heavy manual efforts. Therefore, we propose a totaly unsupervised method USPL to segment the object in videos. Another fact is that visual events are usually accompanied by sounds in our daily lives. There are a few tries which construct a model to learn the correspondence between visual scene and the sound as the self-supervised training strategy. These methods also find that after self-supervised training, model has the ability to locate the sounding related objects by cross modal attention.
>
> However, the ability of sounding localization learnt by self-supervised training seems not strong enough. Our USPL is based on the correspondence self-supervised training and considers the learnt sounding map as a initial sounding area. Then USPL applys a series of refine steps to get pixel-level sounding map results.
>
> The most important step of our refine steps is SMR module which employs the visual semantic affinity learning to explore inter-pixel relations of adjacent coordinate features. It contributes to recovering the boundary of coarse sounding maps, obtaining fine sounding maps. In short, the SMR is a kind of affinity methods which have been used to enhance the quality of semantic segmentation. For example, in [1], CNNs for semantic segmentation are incorporated with a differentiable module computing a semantic affinity matrix of pixels, and trained in an end-to-end manner with full supervision.  In our SMR, we predicts a high-dimensional embedding vector for each pixel, and the affinity between a pair of pixels is defined as the similarity between their embedding vectors.  We use the learnt affinity to propagate coarse sounding map to discover the entire instance areas with accurate boundaries.
>
>
>
> [1] G. Bertasius, L. Torresani, S. X. Yu, and J. Shi. Convolutional random walk networks for semantic image segmentation. In CVPR, 2017
>
>
>
> **Q2** : Have you attempted to leverage the temporal information between frames in your analysis? Could this approach provide any benefits?
>
> **Answers :** Thanks very much for the question.
>
> To leverage the temporal information between frames, we here realize it by replacing the visual backbone from ResNet to C3D which extracts temporal feature. On the AVSBench-S4 validation dataset, our model achieves 0.37% mIoU and 0.06 F-score improvement after employing temporal input as shown in Table 1. This means temporal information is very explorative for our method and we believe temporal information will be a great help to our method after more tries.
>
> **Table 1. Analysis importing temporal information to the proposed USPL. The results are inference procedure on validation set of AVSBench-S4.**
>
> | Methods                   | mIoU(%)$\uparrow$ | F-score$\uparrow$ |
> | ------------------------- | ----------------- | ----------------- |
> | USPL w/ sigle-frame input | 45.41             | 0.565             |
> | USPL w/ sequence input    | 45.78             | 0.571             |
>
>
>
> **Q3** : Have you assessed the contribution of audio to the model? How does the performance vary when audio information is not provided? Can the model still achieve satisfactory performance by merely predicting the mask of the salient object in the image?
>
> **Answers :** Thanks very much for the question.
>
> During the training phase, audio modality serves as a core requirements for our method. As an unsupervised approach, USPL learns the correspondence between visual scene and the sound as a self-supervised training strategy. The correspondence learning of audio and visual process is so much essential. If without it, our USPL can't produce the initial sounding maps and also can't do the next refine processes. Therefore, our USPL must have both audio and visual modalities to work in training and these correspondence two modalities are naturally exist in video data.
>
> During the evaluation phase, audio is not the necessary modality for our method. Because in our evaluation stage, we only use SPS module, and audio modality only participants in the cross-modal fusion to improve the performance. In our ablation experiment, without audio fusion in SPS, the mIoU on AVSBench-S4 validation and test subset only drops 0.05% and  1.46% respectively. This proves that our model still achieves satisfactory performance by merely predicting the mask of the salient object in the image.
>
> Table 2 which analyses the audio modality contribution to the proposed USPL is listed here:
>
> **Table 2. Analysis the audio modality contribution to the proposed USPL. The results are inference procedure on both validation and test subset of AVSBench-S4.**
>
> | mIoU(%)$\uparrow$     | *val* | *test* |
> | --------------------- | ----- | ------ |
> | evaluation w/o audio  | 45.37 | 45.73  |
> | evaluation w/   audio | 45.41 | 47.19  |

---

### Official Review · Reviewer_c1YV · 2023-08-05

**Typos Grammar Style And Presentation Improvements:** 1. In the caption of Fig. 2, SOS shou…
**Soundness:** 4

**Excitement:**

4: Strong: This paper deepens the understanding of some phenomenon or lowers the barriers to an existing research direction.

**Paper Topic And Main Contributions:**

This paper focuses on a novel and interesting task, namely, Unsupervised Sounding Pixel Learning (USPL). The authors proposed a method with a mask augmentation based multi-instance contrastive learning module, an Unsupervised Sounding Map Refinement (SMR) module, and a Sounding Pixel Segmentation (SPS) module. Experiments on public datasets show that the proposed method considerably outperforms current systems and achieves SOTA.

**Questions For The Authors:**

To me, the motivation of introduction of the SPS module remains unclear. What do you mean by ‘Although the SMR produces fine sounding maps, it is too heavy for evaluation’? More explanation may be needed.

**Reasons To Accept:**

1. To my knowledge, the task of Unsupervised Sounding Pixel Learning (USPL) is first explored by this paper. This is a novel, interesting and challenging task which may attract more researchers.
2. Experiments on public datasets show that the proposed method considerably outperforms current systems and achieves SOTA, demonstrating the effectiveness of the paper.

**Reasons To Reject:**

1. I wonder whether the topic of this paper is suitable for EMNLP, since the paper focuses on sounding pixel location, which has nothing to do with NLP.
2. To me, the motivation of introduction of the SPS module remains unclear. What do you mean by ‘Although the SMR produces fine sounding maps, it is too heavy for evaluation’? More explanation may be needed.

**Reproducibility:**

4: Could mostly reproduce the results, but there may be some variation because of sample variance or minor variations in their interpretation of the protocol or method.

**Reviewer Confidence:**

3: Pretty sure, but there's a chance I missed something. Although I have a good feel for this area in general, I did not carefully check the paper's details, e.g., the math, experimental design, or novelty.

---

> ### Author Rebuttal · Authors · 2023-08-28
>
> Thank you for your insightful feedback on our paper. We sincerely appreciate your time and effort in reviewing our work. We would like to address your concern with the following points:
>
> **Q1** : To me, the motivation of introduction of the SPS module remains unclear. What do you mean by ‘Although the SMR produces fine sounding maps, it is too heavy for evaluation’? More explanation may be needed.
>
> **Answers :** Thanks very much for your valuable suggestion.
>
> Here we discuss the model computation and memory costs to explain the motivation of SPS as shown in Table 1.
>
> In our proposed method, the SMR and SPS modules play a crucial role. SMR module refines the sounding maps by leveraging visual semantic affinity, resulting in fine-grained sounding maps. However, it is noted that SMR exhibits low efficiency. Therefore, SPS module serves as a lightweight evaluation module. We calculate the *parameters of model (Params)*,  *floating point operations (FLOPS)*  and *frames per second (FPS)* to evaluate methods' efficiency. We find that using SPS to bypass SMR could slightly reduce space complexity and significantly decreases time complexity, which leading to better performance.
>
> **Table 1. Analysis computation and memory cost of USPL evaluation. The results are inference procedure on validation set of AVSBench-S4.**
>
> |            | mIoU(%)$\uparrow$ | Params(M)$\downarrow$ | Flops(G)$\downarrow$ | FPS$\uparrow$ |
> | ---------- | ----------------- | --------------------- | -------------------- | ------------- |
> | USPL (SMR) | 44.97             | 69.64                 | 40.85                | 4.34          |
> | USPL (SPS) | 45.41             | 68.12                 | 38.61                | 9.94          |
>
>
>
> **Q2**  : In the caption of Fig. 2, SOS should be SPS.
>
> **Answers:** Thanks very much for pointing out our issue.
>
> We sincerely apologize for mistakenly writing "SPS" to "SOS" in the caption of Fig. 2. We have thoroughly check and correct it.
>
>
>
> **Q3** : In eq 11 and other places, why mmicl and seg are italic while aff is not?
>
> **Answers:** Thank you very much for pointing out our problem.
>
> We sincerely apologize for mistakenly font using. We have thoroughly check and correct it.

---

### Official Review · Reviewer_A2BU · 2023-08-10

**Soundness:** 4

**Excitement:**

4: Strong: This paper deepens the understanding of some phenomenon or lowers the barriers to an existing research direction.

**Paper Topic And Main Contributions:**

This paper aims to generate precise pixel-level object maps from videos under the unsupervised setting. Specifically, the authors propose a two-step approach: initially aligning audio-visual features coarsely using the MMICL module, followed by refining the coarse maps using an unsupervised SMR module. Additionally, they incorporate a sounding pixel segmentation (SPS) module for lightweight evaluation and improved performance. The experimental results demonstrate the effectiveness of the proposed methods.

**Reasons To Accept:**

1. The proposed Sounding Map Refinement (SMR) module is novel and reasonable to me.

2. The experimental results indicate that the proposed method outperforms previous unsupervised methods.

3. The paper is well-written and easy to follow.

**Reasons To Reject:**

1. The efficiency of the SMR module appears to be low. Although the authors suggest using SPS to bypass the SMR during the evaluation stage, there is no corresponding discussion in the experimental section. I hope to see a comparison of the time and space complexity with and without SPS during evaluation.

2. Since the SMR module significantly contributes to the performance improvement, has there been any exploration of the hyperparameters used in Eq. 5, 7, and 8?

**Reproducibility:**

4: Could mostly reproduce the results, but there may be some variation because of sample variance or minor variations in their interpretation of the protocol or method.

**Reviewer Confidence:**

4: Quite sure. I tried to check the important points carefully. It's unlikely, though conceivable, that I missed something that should affect my ratings.

---

> ### Author Rebuttal · Authors · 2023-08-28
>
> We express our gratitude for your insightful feedback on our paper. We sincerely appreciate your time and effort in reviewing our work. In response to your concerns, we would like to provide the following clarifications:
>
> **Q1** : The efficiency of the SMR module appears to be low. Although the authors suggest using SPS to bypass the SMR during the evaluation stage, there is no corresponding discussion in the experimental section. I hope to see a comparison of the time and space complexity with and without SPS during evaluation.
>
> **Answers :** Thanks very much for your valuable suggestion.
>
> Here we discuss the model computation and memory costs as shown in Table 1.
>
> In our proposed method, the SMR and SPS modules play a crucial role. SMR module refines the sounding maps by leveraging visual semantic affinity, resulting in fine-grained sounding maps. However, it is noted that SMR exhibits low efficiency. Besides, SPS module serves as a lightweight evaluation module. We calculate the *parameters of model (Params)*,  *floating point operations (FLOPS)*  and *frames per second (FPS)* to evaluate methods' efficiency. We find that using SPS to bypass SMR may slightly reduce space complexity and significantly decreases time complexity, which leading to better performance.
>
> **Table 1. Analysis computation and memory cost of USPL evaluation. The results are inference procedure on validation set of AVSBench-S4.**
>
> |            | mIoU(%)$\uparrow$ | Params(M)$\downarrow$ | FLOPS(G)$\downarrow$ | FPS$\uparrow$ |
> | :--------- | ----------------- | --------------------- | :------------------- | ------------- |
> | USPL (SMR) | 44.97             | 69.64                 | 40.85                | 4.34          |
> | USPL (SPS) | 45.41             | 68.12                 | 38.61                | 9.94          |
>
>
>
> **Q2** : Since the SMR module significantly contributes to the performance improvement, has there been any exploration of the hyperparameters used in Eq. 5, 7, and 8?
>
> **Answers :** Thanks very much for the question.
>
> SMR(Sounding Map Refinement) module contributes significantly to our sounding object localization performance improvement. Here we will explore the hyper-parameters involved in SMR as shown in Table 2. We first explore the hyper-parameters in Eq. 5, specifically focusing on the two background thresholds $\beta_l$ and $\beta_h$ which control the Tri-map division.  We set a list of hyper-parameters finding that $\beta_l$ = 0.4 and $\beta_h$ = 0.55 are best in Eq. 5, respectively.
>
> **Table 2. Evaluation on the background thresholds $\beta_h$, $\beta_l$ of Tri-map generation on AVSBench-S4 validation.**
>
> | $\beta_h$ | $\beta_l$ | mIoU(%)$\uparrow$ | F-score$\uparrow$ |
> | --------- | --------- | ----------------- | ----------------- |
> | 0.7       | 0.6       | 30.69             | 0.473             |
> | 0.6       | 0.5       | 37.94             | 0.527             |
> | **0.5**   | **0.4**   | **45.41**         | **0.565**         |
> | 0.4       | 0.3       | 41.19             | 0.492             |
>
> After exploring the hyper-parameters in Eq. 5, we also delve into the hyper-parameters of Eq. 7 and Eq. 8, which significantly impact sounding map refinement. The $\beta$ parameter in Eq. 7 serves as a hyper-parameter for ignoring trivial affinity values in the semantic affinity matrix, while the $t$ parameter in Eq. 8 controls the iterations of the semantic transition matrix. Our parameters search through grid-based evaluation reveals that setting $\beta$ = 10 in Eq. 7 and $t$ = 8 in Eq. 8 yields optimal results for our model as shown in Table 3.
>
> **Table 3. Evaluation on the hyper-parameters $\beta$, $t$ of the sounding map refinement on AVSBench-S4 validation, reported in mIoU.**
>
> |         |        |       | $t$   |           |       |
> | ------- | ------ | ----- | ----- | --------- | ----- |
> |         |        | 1     | 4     | **8**     | 12    |
> |         | 1      | 39.53 | 34.26 | 15.19     | 15.08 |
> | $\beta$ | 5      | 40.49 | 43.68 | 36.61     | 13.18 |
> |         | **10** | 40.07 | 42.51 | **45.41** | 24.03 |
> |         | 15     | 39.75 | 41.65 | 44.95     | 37.77 |

---

### Official Review · Reviewer_3pBn · 2023-08-12

**Typos Grammar Style And Presentation Improvements:** Line 521 mentions "Fig. 3", and I thi…
**Soundness:** 4

**Excitement:**

4: Strong: This paper deepens the understanding of some phenomenon or lowers the barriers to an existing research direction.

**Paper Topic And Main Contributions:**

In this paper authors propose Unsupervised Sounding Pixel Learning (USPL) to improve cross-model pixel-level sounding source localization. USPL consists of modules for mask augmentation based multi-instance contrastive learning (MMICL), Sounding Map Refinement (SMR), Sounding Pixel Segmentation (SPS) etc. Experimental results show the effectiveness of the proposed approach.

**Questions For The Authors:**

See the questions listed above.

**Reasons To Accept:**

It's with broad interests to investigate pixel sound source location in an unsupervised manner, which mitigates the challenges of obtaining large volume of annotations. Overall, this paper clearly presents the proposed methodology, with well-designed experimental design and ablation studies.

**Reasons To Reject:**

I think more details could be added to strengthen the experimental session, including:
1. How to determine the values of hyper-parameters, e.g. as described in Section 4.1.
2. For methods listed in Table 2, it helps to include computational and memory costs.
3. For the ablation study summarized in Table 3, why is the test partition (not the dev partition) of AVSBench-S4 used?



**Reproducibility:**

4: Could mostly reproduce the results, but there may be some variation because of sample variance or minor variations in their interpretation of the protocol or method.

**Reviewer Confidence:**

3: Pretty sure, but there's a chance I missed something. Although I have a good feel for this area in general, I did not carefully check the paper's details, e.g., the math, experimental design, or novelty.

---

> ### Author Rebuttal · Authors · 2023-08-28
>
> Thank you for your insightful feedback on our paper. We sincerely appreciate your time and effort in reviewing our work. We would like to address your concern with the following points:
>
> **Q1** : How to determine the values of hyper-parameters, e.g. as described in Section 4.1.
>
> **Answers** : Thanks very much for your valuable feedback.
>
> In Section 4.1, we list some hyper-parameter settings of USPL . To ensure the optimal performance of the algorithm, we conduct extensive experiments on the AVSBench-S4 dataset, where we explore alternative hyper-parameter combinations. The ablation studies are conducted on AVSBench-S4 train subset and evalated on AVSBench-S4 validation subset.
>
> We firstly ablate the hyper-parameters in MMICL module. Specifically, we focus on the hyper-parameter temperature  $\tau$ in Eq. 2, which serves a similar role as the temperature in contrastive learning. This parameter controls the shape of the distribution and plays a crucial role in distinguishing negative samples. Through extensive experiments as shown in Table 1, we find that setting $\tau$ to 0.07 yields the best result.
>
> **Table 1. Evaluation on the temperature parameter $\tau$ of MMICL loss on AVSBench-S4 validation set.**
>
> | $\tau$   | mIoU(%)$\uparrow$ | F-score$\uparrow$ |
> | -------- | ----------------- | ----------------- |
> | 0.03     | 44.67             | 0.544             |
> | **0.07** | **45.41**         | **0.565**         |
> | 0.10     | 41.75             | 0.539             |
> | 0.13     | 39.19             | 0.517             |
>
> Then, we investigate the impact of the drop probability $\delta$ in the random mask augmentation layer as shown in Table 2. This parameter controls the probability of dropping visual features during training, which makes model more robustness. We perform a series of settings and compare the results to the case where no random mask augmentation is used, i.e., $\delta = 0$. It is observed that using random mask always yields better performance. Moreover, when $\delta = 0.5$, the model achieves its optimal performance. It is also important to note that excessive drop probability can be detrimental to the model's performance.
>
> **Table 2. Evaluation on the drop probability $\delta$  of Random mask augmentation layer on AVSBench-S4 validation set.**
>
> | $\delta$ | mIoU(%)$\uparrow$ | F-score$\uparrow$ |
> | -------- | ----------------- | ----------------- |
> | 0        | 43.07             | 0.545             |
> | 0.1      | 44.87             | 0.564             |
> | 0.3      | 45.12             | 0.561             |
> | **0.5**  | **45.41**         | **0.565**         |
> | 0.7      | 44.63             | 0.554             |
> | 0.9      | 13.49             | 0.223             |
>
> Subsequently, we conduct ablation studies on the hyper-parameter in the SMR module as shown in Table 3. The SMR module is responsible for refining the object shape information from coarse sounding maps, which significantly contributes to the improvement of our sounding object localization performance. The parameter $r$ controls the radius of the affinity matrix generation, which determines the scope of interest neighborhood around each point in the coarse sounding maps. In our experiments, we find that setting $r=7$ yields the best results. It's noted that excessive values of $r$ not only consume more computational resources but also generate unnecessary relations that can adversely affect the final outcome.
>
> **Table 3. Evaluation on the radius $r$ of affinity matrix generation on AVSBench-S4 validation set.**
>
> | radius $r$ | mIoU(%)$\uparrow$ | F-score$\uparrow$ |
> | ---------- | ----------------- | ----------------- |
> | 3          | 42.27             | 0.531             |
> | 5          | 43.77             | 0.551             |
> | **7**      | **45.41**         | **0.565**         |
> | 9          | 39.59             | 0.502             |
>
> We explore the hyper-parameter in Eq. 5, specifically focusing on the two background thresholds $\beta_l$ and $\beta_h$ which control the Tri-map division. Through extensive experiments in Table 4, we establish that setting $\beta_l$ to 0.4 and $\beta_h$ to 0.55 yields optimal results in Eq. 5, respectively.
>
> **Table 4. Evaluation on the background thresholds $\beta_h$, $\beta_l$ of Tri-map generation on AVSBench-S4 validation set.**
>
>
> | $\beta_h$ | $\beta_l$ | mIoU(%)$\uparrow$ | F-score$\uparrow$ |
> | --------- | --------- | ----------------- | ----------------- |
> | 0.75      | 0.6       | 30.69             | 0.473             |
> | 0.65      | 0.5       | 37.94             | 0.527             |
> | **0.55**  | **0.4**   | **45.41**         | **0.565**         |
> | 0.45      | 0.3       | 41.19             | 0.492             |
>
> In Table 5, we explore the hyper-parameters in Eq. 5, we also delve into the hyper-parameters of Eq. 7 and Eq. 8, which significantly impact sounding map refinement. The $\beta$ in Eq. 7 is for ignoring trivial affinity values in the semantic affinity matrix, while the $t$ in Eq. 8 controls the iterations of the semantic transition matrix. Our parameters search through grid-based evaluation reveals that setting $\beta$ = 10 and $t$ = 8 yields optimal results for our model.
>
> **Table 5. Evaluation on the hyper-parameters $\beta$, $t$ of the sounding map refinement on AVSBench-S4 validation set, reported in mIoU.**
>
> |         |        |       | $t$   |           |       |
> | ------- | ------ | ----- | ----- | --------- | ----- |
> |         |        | 1     | 4     | **8**     | 12    |
> |         | 1      | 39.53 | 34.26 | 15.19     | 15.08 |
> | $\beta$ | 5      | 40.49 | 43.68 | 36.61     | 13.18 |
> |         | **10** | 40.07 | 42.51 | **45.41** | 24.03 |
> |         | 15     | 39.75 | 41.65 | 44.95     | 37.77 |
>
>
>
> **Q2** : For methods listed in Table 2(in our paper), it helps to include computational and memory costs.
>
> **Answers :** Thanks very much for your valuable suggestion.
>
> Here we discuss the model computation and memory costs as shown in Table 6.
>
> We calculate the *parameters of model (Params)*,  *floating point operations (FLOPS)*  and *frames per second (FPS)* to evaluate methods' efficiency. The methods listed in Table2(in our paper) are a series of similar methods and we choose the representative work EZ-VSL which also our baseline, showing on the ablation experiment result table.
>
> Comparing to the baseline method, our USPL exhibits lower efficiency both in time and space complexity. However, we have noticed the computational efficiency problem and SPS module is proposed for boosting the inference speed. What's more, as part of future work, we aim to improve the efficiency of our method.
>
> **Table 6. Analysis computation and memory cost of USPL evaluation. The results are inference procedure on validation set of AVSBench-S4.**
>
> |                   | mIoU(%)$\uparrow$ | Params(M)$\downarrow$ | FLOPS(G)$\downarrow$ | FPS$\uparrow$ |
> | :---------------- | ----------------- | --------------------- | :------------------- | ------------- |
> | EZ-VSL (baseline) | 33.85             | 24.14                 | 6.46                 | 15.45         |
> | USPL (SMR)        | 44.97             | 69.64                 | 40.85                | 4.34          |
> | USPL (SPS)        | 45.41             | 68.12                 | 38.61                | 9.94          |
>
>
>
> **Q3** : For the ablation study summarized in Table 3(in our paper), why is the test partition (not the dev partition) of AVSBench-S4 used?
>
> **Answers :** Thanks very much for the question.
>
> In our paper, Tabel 7 shows the ablation study of various modules in USPL. To keep consistent with previous works, we report the experiments on test subset of AVSBench-S4.  We also evaluate each module on validation subset and we strictly comply with the machine learning training protocol, which means test subset is only used for final test and validation subset is used for hyper-parameter tuning.
>
> We will report both the validation subset and the test subset results of this ablation study in the final version of our paper.
>
> **Table 7. Ablation study of different modules of the proposed USPL in mIoU. The results are inference procedure on both validation and test subset of AVSBench-S4.**
>
> | Methods          | MMICL | SMR  | SPS  | *val*     | *test*    |
> | ---------------- | ----- | ---- | ---- | --------- | --------- |
> | EZ-VSL(Baseline) |       |      |      | 33.85     | 34.01     |
> |                  | √     |      |      | 34.09     | 35.52     |
> | USPL(Ours)       | √     | √    |      | 43.72     | 44.01     |
> |                  | √     | √    | √    | **45.41** | **47.19** |
>
>
>
> **Q4 :** Line 521 mentions "Fig. 3", and I think authors meant "Fig. 4" there?
>
> **Answers :** Thanks very much for pointing out our issue.
>
> We sincerely apologize for mistakenly writing "Fig. 4" to "Fig. 3". We have thoroughly check and correct it.

---

### Meta-Review · Area_Chair_wvJb · 2023-09-15

**Recommendation:** 4

**Metareview:**

This paper studies unsupervised sound source localization from multimodal input (images of objects that make sound, along with the corresponding sound waveforms they produce).

Pros (from reviewers):
The paper studies an interesting and well-motivated problem
The presentation of the experiments in the paper is clear
The experimental results demonstrate that the proposed method outperforms previous models
Some of the model components are technically novel (Sounding Map Refinement)

Cons (from reviewers):
The work may or may not be a good fit for EMNLP (it does deal with multimodality and sound, but not with language)
The motivation and reasoning behind some of the model components (and why they work) is not adequately explained
The experiments do not explore computational or memory costs
The paper did not fully explore the hyperparameter tuning of its proposed components

---

### Decision · Program_Chairs · 2023-10-07

**Decision:**

Accept-Main

**Comment:**

This paper studies unsupervised sound source localization from multimodal input (images of objects that make sound, along with the corresponding sound waveforms they produce).

Pros (from reviewers):
The paper studies an interesting and well-motivated problem
The presentation of the experiments in the paper is clear
The experimental results demonstrate that the proposed method outperforms previous models
Some of the model components are technically novel (Sounding Map Refinement)

Cons (from reviewers):
The work may or may not be a good fit for EMNLP (it does deal with multimodality and sound, but not with language)
The motivation and reasoning behind some of the model components (and why they work) is not adequately explained
The experiments do not explore computational or memory costs
The paper did not fully explore the hyperparameter tuning of its proposed components